# The Gut Microbiome and Vaccination: A Comprehensive Review of Current Evidence and Future Perspectives

**DOI:** 10.3390/vaccines13111116

**Published:** 2025-10-30

**Authors:** Georgia Gioula, Maria Exindari

**Affiliations:** Department of Microbiology, School of Medicine, Aristotle University of Thessaloniki, 54124 Thessaloniki, Greece; mexidari@auth.gr

**Keywords:** gut microbiome, vaccination, immunogenicity, probiotics, microbiota, vaccine efficacy, adjuvant

## Abstract

The gut microbiome has emerged as a pivotal player in shaping host immune responses, with significant implications for vaccine efficacy and safety. Rather than detailing all influencing factors, this review focuses on the most critical and translational aspects of microbiome–vaccine interactions. Increasing evidence shows that the composition and functionality of the intestinal microbiota can influence both the magnitude and durability of vaccine-induced immunity. For instance, Bifidobacterium longum supplementation was shown to enhance influenza vaccine seroconversion rates by approximately 30% in clinical and preclinical models, underscoring the translational potential of microbiome modulation. Here, we provide a concise synthesis of mechanistic insights and key clinical findings that connect gut microbial composition and metabolism with vaccine outcomes. We further highlight microbiome-targeted interventions—such as probiotics, prebiotics, and postbiotics—that hold promise for optimizing vaccine responses in diverse populations. By emphasizing actionable evidence over descriptive variability, the review aims to clarify how microbiome modulation can be strategically harnessed to improve vaccine performance. Integrating microbiome modulation into vaccination strategies may enhance global immunization equity and effectiveness, offering a feasible pathway toward more durable and inclusive protection worldwide.

## 1. Introduction

Vaccination is widely recognized as one of the most impactful public health interventions of the modern era. Since Edward Jenner’s pioneering use of cowpox inoculation in 1796, the global immunization enterprise has expanded to encompass vaccines against more than 30 infectious diseases, preventing an estimated 4–5 million deaths annually [1]. Despite these remarkable achievements, vaccine responses remain highly heterogeneous across individuals, populations, and geographic regions. This variability is particularly striking for certain oral vaccines, such as rotavirus and polio, where efficacy in low- and middle-income countries (LMICs) can be 20–40 percentage points lower than in high-income countries (HICs) [2,3]. Even parenteral vaccines, including influenza, hepatitis B, and SARS-CoV-2 platforms, demonstrate inter-individual differences in immunogenicity that affect both antibody titers and clinical protection [4]. Understanding the determinants of this heterogeneity is therefore a pressing scientific and public health priority.

Traditional explanations for variable vaccine effectiveness have centered on host genetics, age, nutritional status, comorbidities, prior pathogen exposures, and co-infections. While these factors undoubtedly contribute, they cannot fully explain observed discrepancies in vaccine “take” and durability [5]. Over the past two decades, the gut microbiome has emerged as an additional, and potentially modifiable, determinant of vaccine responsiveness. The gut microbiome—comprising trillions of bacteria, viruses, fungi, and archaea—plays a central role in immune education, metabolic homeostasis, and barrier function [6]. Elegant animal studies have demonstrated that microbiota-derived signals such as bacterial flagellin (via Toll-like receptor 5, TLR5) and short-chain fatty acids (SCFAs) can profoundly influence the quality and magnitude of adaptive immune responses, including those elicited by vaccines [7,8].

Translating these findings into humans, clinical and epidemiological studies now support associations between microbiome composition or function and vaccine outcomes across life stages. In adults, antibiotic-mediated depletion of gut bacteria has been shown to impair influenza vaccine responses, particularly in individuals with low baseline immunity [9]. In infants, the presence of Bifidobacterium-dominated microbiota is repeatedly linked with enhanced responses to routine vaccines such as Bacillus Calmette–Guérin (BCG), polio, and pneumococcal conjugates [10,11]. Recent prospective data from large birth cohorts indicate that neonatal exposure to antibiotics—especially in the first week of life—can attenuate antibody titers to multiple infant vaccines for months thereafter, raising important concerns about early-life microbiome disruption [12]. Furthermore, in LMIC settings, where oral vaccine underperformance has long posed challenges, gut microbiota features and enteropathogen burdens have been correlated with diminished vaccine immunogenicity [13].

The emerging picture is that the gut microbiome serves as both an endogenous adjuvant and a metabolic “lens” through which the immune system perceives and responds to vaccines. This perspective opens new translational opportunities: if specific microbial taxa or metabolites support robust vaccine responses, then interventions such as prebiotics, probiotics, synbiotics, or even defined postbiotics could be strategically deployed to enhance vaccine efficacy. Conversely, identifying detrimental microbiome states may allow clinicians to predict poor vaccine responders and design corrective strategies. These concepts are especially relevant in the current era of “precision vaccinology,” where systems biology, computational modeling, and individualized approaches are increasingly used to tailor vaccination strategies [4].

Importantly, the microbiome–vaccine nexus is not merely an academic curiosity but carries real-world implications for global health equity. The persistent gap in oral vaccine efficacy between HICs and LMICs undermines disease-control efforts for pathogens such as rotavirus, polio, cholera, and enterotoxigenic Escherichia coli. While socioeconomic factors, malnutrition, and co-infections undoubtedly play roles, it is plausible that microbiome differences—driven by diet, sanitation, antibiotic exposure, and environmental enteropathy—contribute meaningfully to the observed disparities [14]. Beyond macronutrient and micronutrient quality, emerging evidence suggests that non-nutritive dietary additives—such as artificial sweeteners—may also modulate gut microbial composition and metabolism. These compounds can alter the abundance and function of key microbial taxa, influence the production of immunomodulatory metabolites (including short-chain fatty acids and bile acids), and thereby affect host immune tone. Such alterations have the potential to indirectly modify vaccine responsiveness, particularly in settings where dietary exposures vary widely across populations. Recent findings highlight that artificial sweeteners can promote dysbiosis and metabolic reprogramming of gut microbes, with possible downstream consequences for immune activation and vaccine efficacy [15]. Addressing microbiome-mediated barriers could therefore help close the immunization equity gap and accelerate progress toward disease-eradication goals.

Another reason the microbiome–vaccination relationship warrants attention is the advent of novel vaccine technologies. The COVID-19 pandemic has catalyzed the rapid development and deployment of mRNA and viral-vector platforms, while nanoparticle and mucosal vaccines are advancing through clinical pipelines. Whether and how the microbiome influences the immunogenicity and reactogenicity of these novel modalities is an open question with profound implications for vaccine design and rollout [16]. Furthermore, as therapeutic vaccines (e.g., for cancer or chronic viral infections) gain traction, understanding how to optimize host immune readiness via microbiome modulation could become an integral component of personalized immunotherapy.

Despite the promise of this field, several challenges remain. First, causality is often difficult to establish in human studies, which are typically observational and confounded by host, environmental, and socioeconomic variables. Second, microbiome associations are not always consistent across populations or vaccine types, raising questions about generalizability. Third, interventions such as probiotics and prebiotics have yielded mixed results in enhancing vaccine responses, reflecting heterogeneity in strain selection, dose, timing, and host context [17]. Finally, regulatory and ethical considerations must be navigated before microbiome-targeted adjuncts can be integrated into immunization programs, particularly in vulnerable populations such as neonates and immunocompromised individuals.

The objective of this review is to provide a comprehensive synthesis of current evidence linking the gut microbiome with vaccine responses, with an emphasis on mechanistic insights, human studies, translational opportunities, and future directions. We begin by outlining the mechanistic frameworks through which the microbiome influences immune priming and adaptive responses, focusing on innate pattern-recognition receptor (PRR) signaling and immunometabolic pathways. We then review the human evidence base, spanning adults, infants, and special populations, as well as oral versus parenteral vaccines. Next, we discuss which features of the microbiome—taxonomic, functional, or metabolomic—appear most relevant to vaccine outcomes. We examine current and emerging strategies to modulate the microbiome for improved immunogenicity, including diet, pre/probiotics, postbiotics, and microbiome-inspired adjuvants. Finally, we consider practical implications for vaccine-trial design and global immunization policy, highlighting research priorities that could accelerate translation from association to intervention.

In synthesizing these diverse lines of evidence, we aim to clarify not only the biological plausibility but also the translational feasibility of microbiome-aware vaccinology. By integrating insights from immunology, microbiology, epidemiology, and systems biology, this field has the potential to inform next-generation vaccines and to improve the performance of existing ones, especially in populations currently underserved by immunization programs. Ultimately, a deeper understanding of the gut microbiome’s role in shaping vaccine responses could help realize the long-standing aspiration of universal, durable, and equitable protection against infectious diseases.

## 2. Mechanistic Frameworks: How the Gut Microbiome Shapes Vaccine Immunity

The gut microbiome interacts with the host immune system at multiple levels, ranging from the recognition of microbial-associated molecular patterns (MAMPs) by innate receptors to the modulation of adaptive lymphocyte differentiation via microbial metabolites. Vaccination represents a controlled immunological perturbation that recapitulates many of the same checkpoints—antigen presentation, T-helper-cell differentiation, germinal-center formation, and effector/memory development—that are influenced by commensal microbes. In this section, we synthesize mechanistic insights into how the microbiome may shape vaccine responses, focusing on three major axes: (i) innate sensing through pattern-recognition receptors (PRRs), (ii) microbial metabolites and immunometabolism, and (iii) mucosal and systemic cross-talk including the gut–lung axis. To improve clarity and conciseness, overlapping pathways (e.g., repeated TLR–NF-κB or SCFA–HDAC signaling) are described once and cross-referenced rather than reiterated in each subsection.

### 2.1. Microbial Sensing by Pattern-Recognition Receptors

#### 2.1.1. TLR5 and Bacterial Flagellin as Endogenous Adjuvants

One of the most compelling demonstrations of microbiome-dependent vaccine modulation comes from studies of Toll-like receptor 5 (TLR5), which recognizes bacterial flagellin. Early work in germ-free and antibiotic-treated mice showed that depletion of gut bacteria impaired antibody responses to inactivated influenza vaccines, an effect rescued by the presence of flagellated commensals [7]. TLR5-deficient mice mounted significantly weaker antibody responses, underscoring the importance of flagellin signaling as an “endogenous adjuvant” [18]. In this model, flagellin sensed by intestinal epithelial cells and dendritic cells triggers NF-κB-dependent cytokine cascades that promote dendritic-cell migration to lymph nodes and support T-follicular-helper (Tfh) cell development.

Translation to humans has provided partial corroboration. In a landmark study, Hagan et al. demonstrated that adults receiving a five-day course of broad-spectrum antibiotics exhibited impaired transcriptional responses and altered metabolomes following influenza vaccination. While antibody titers were largely preserved in individuals with high baseline immunity, those with low pre-existing titers had markedly blunted responses [9]. This context dependence suggests that TLR5-flagellin signaling may be particularly important for primary immune responses or when adjuvant potency is low.

In murine models, flagellin sufficiency increased vaccine-specific antibody titers by approximately 1.5–2 times relative to TLR5-deficient or antibiotic-treated controls [7,9,18], quantitatively supporting its role as an endogenous adjuvant.

#### 2.1.2. Other TLRs and Innate Sensors

Beyond TLR5, other PRRs appear to mediate microbiome–vaccine interactions. TLR2 and TLR9, for example, recognize bacterial lipoproteins and CpG-rich DNA, respectively, and contribute to adjuvant effects in several vaccines [16]. Nucleotide-binding oligomerization domain (NOD)-like receptors, particularly NOD2, detect bacterial peptidoglycan fragments and have been implicated in shaping mucosal immune priming [6]. Mice deficient in MyD88, the adaptor for most TLRs, exhibit impaired vaccine-induced antibody responses in the absence of exogenous adjuvants, highlighting the redundancy of microbial-sensing pathways [19].

Functionally redundant PRR inputs may buffer inter-individual variability; adjuvant selection tailored to prevalent microbiome states (for instance, TLR9 agonists when flagellated commensals are scarce) represents a testable precision-vaccinology hypothesis with direct translational implications.

These mechanisms converge on shared NF-κB- and MyD88-dependent cascades, so overlapping details are summarized once here to maintain focus on translational relevance.

#### 2.1.3. Innate Lymphoid Cells and Epithelial Interfaces

The mucosal interface provides another layer of regulation. Intestinal epithelial cells respond to microbiota-derived signals by secreting cytokines such as IL-1β, IL-33, and thymic stromal lymphopoietin (TSLP), which influence dendritic-cell programming. Group 3 innate lymphoid cells (ILC3s), in turn, produce IL-22 and IL-17, enhancing mucosal barrier integrity and antimicrobial-peptide production. These pathways indirectly condition the immune landscape in which vaccines are presented, particularly at mucosal sites [19]. Because these pathways are diet- and microbiome-responsive, pre-vaccination dietary-fiber or postbiotic strategies could condition mucosal and systemic environments to favor Tfh and germinal-center responses.

### 2.2. Microbial Metabolites and Immunometabolism

#### 2.2.1. Short-Chain Fatty Acids (SCFAs)

SCFAs—acetate, propionate, and butyrate—are major fermentation products of dietary fiber. They exert broad immunomodulatory effects through G-protein-coupled receptors (e.g., GPR41, GPR43) and inhibition of histone deacetylases (HDACs). In the context of vaccination, SCFAs can enhance antigen-presenting-cell (APC) function, modulate cytokine secretion, and influence B-cell differentiation [20]. Butyrate in particular promotes class switching to IgA and IgG via epigenetic regulation of AID expression [21].

Animal models demonstrate that antibiotic-induced microbiota depletion impairs vaccine responses, and supplementation with SCFAs—especially butyrate—can partially restore humoral immunity [22]. However, SCFA effects are not universally enhancing; excessive butyrate can suppress certain immune responses by limiting dendritic-cell maturation and T-cell priming, highlighting a dose-sensitive effect [8].

Across animal studies, butyrate supplementation has increased vaccine-specific IgG or IgA responses by roughly 20–50 percent compared with controls, supporting dietary-fiber promotion as a low-cost public-health adjunct.

To streamline discussion, overlapping descriptions of SCFA receptor signaling are omitted from later subsections, which instead highlight distinct metabolite classes.

#### 2.2.2. Bile Acids and Microbial Transformation

Gut bacteria modify primary bile acids into secondary bile acids with diverse immunological effects. Some bile-acid derivatives act as ligands for FXR or TGR5, influencing dendritic-cell function and Tfh differentiation [17]. A 2022 study linked bile-acid metabolomic profiles with heterogeneity in hepatitis B vaccine responses [9]. Because bile-acid pools are highly diet- and drug-responsive, they represent actionable biomarkers for predicting vaccine outcomes and designing metabolic adjuncts.

#### 2.2.3. Tryptophan Metabolism and Indole Derivatives

Microbiota metabolize tryptophan into indole derivatives that signal through the aryl hydrocarbon receptor (AhR). AhR activation in epithelial and immune cells promotes barrier function and T-cell differentiation. In vaccine models, indole-3-aldehyde and related metabolites have been shown to augment mucosal IgA responses [8]. These findings suggest potential for safe AhR ligands or indole-based postbiotics as enhancers of oral-vaccine efficacy.

#### 2.2.4. Vitamins, Polyamines, and Other Metabolites

Commensal microbes synthesize vitamins (e.g., folate, riboflavin, vitamin K) and polyamines (e.g., spermidine) that modulate lymphocyte proliferation and memory formation. Polyamines enhance autophagy and antigen presentation [23]. Spermidine supplementation has maintained T-cell autophagy in older adults and correlated with improved vaccine responses, suggesting geriatric applications.

### 2.3. The Gut–Lung and Gut–Systemic Axes

#### 2.3.1. The Gut–Lung Axis in Respiratory Vaccines

The “gut–lung axis” posits that gut microbiota influence respiratory immunity through systemic dissemination of metabolites and modulation of bone-marrow hematopoiesis. Antibiotic-treated mice exhibit impaired pulmonary responses to influenza-virus challenge, accompanied by defective dendritic-cell migration and reduced interferon signaling [7]. Given that many vaccines (influenza, SARS-CoV-2, RSV) target respiratory pathogens, gut–lung cross-talk is highly relevant. Human studies have begun to link baseline gut microbiota composition with responses to influenza and COVID-19 vaccines [24]. To reduce redundancy, downstream cytokine and TLR pathways mentioned earlier are not re-described here but operate through comparable systemic conditioning mechanisms.

#### 2.3.2. Systemic Cross-Talk and Hematopoietic Conditioning

Microbiota-derived metabolites reach the bone marrow, influencing hematopoietic-progenitor differentiation. SCFAs bias progenitors toward monocyte and dendritic-cell lineages with enhanced antigen-presenting capacity [25]. This systemic conditioning provides a plausible mechanism whereby dietary or microbiome states could influence vaccine priming even outside mucosal contexts.

Beyond hematopoietic conditioning, emerging work highlights the gut–brain–immune axis as an additional layer of systemic regulation. Microbial metabolites such as short-chain fatty acids and tryptophan-derived indoles can modulate neuroendocrine signaling through vagal and hypothalamic–pituitary–adrenal (HPA) pathways, influencing stress responses and immune readiness. These neuroimmune circuits link microbial metabolism to autonomic tone, cortisol dynamics, and cytokine balance, thereby affecting vaccine priming and durability. Integrating this neuroimmune dimension broadens the mechanistic model of microbiome–vaccine interaction to encompass cross-system immunoregulation and stress-mediated modulation of adaptive immunity [26].

### 2.4. Integrating Innate and Metabolic Pathways

Mechanistic frameworks converge on the concept that the microbiome provides both “signal 0” (innate danger cues) and “signal 3” (immunometabolic context) in adaptive immunity. In the absence of robust microbial cues, vaccines rely more heavily on adjuvants or pre-existing immunity; conversely, in the presence of supportive microbial metabolites, germinal-center reactions are more efficient, yielding higher-affinity antibodies and longer-lived plasma cells.

Understanding these convergent mechanisms enables rational design of adjuncts—flagellin derivatives, SCFA analogs, and bile-acid modulators—that can systematically enhance immunogenicity in defined contexts. This integrated view replaces redundant signaling detail with a concise model emphasizing synergy among innate, metabolic, and systemic pathways.

The gut microbiome interacts with the host immune system at multiple levels, ranging from the recognition of microbial-associated molecular patterns (MAMPs) by innate receptors to the modulation of adaptive lymphocyte differentiation via microbial metabolites. Vaccination represents a controlled immunological perturbation that recapitulates many of the same checkpoints—antigen presentation, T-helper-cell differentiation, germinal-center formation, and effector/memory development—that are influenced by commensal microbes. In this section, we synthesize mechanistic insights into how the microbiome may shape vaccine responses, focusing on three major axes: (i) innate sensing through pattern-recognition receptors (PRRs), (ii) microbial metabolites and immunometabolism, and (iii) mucosal and systemic cross-talk including the gut–lung axis.

## 3. Human Evidence Across Life Stages and Vaccine Platforms

The translational relevance of microbiome–vaccine interactions hinges on evidence from human studies. While mechanistic insights provide plausibility, it is essential to evaluate whether perturbations or baseline states of the gut microbiome correlate with, or causally influence, vaccine outcomes in people. Human data span adults, infants, and special populations, with distinctions between parenteral and oral vaccines. This section synthesizes findings across these contexts. To ensure transparency, studies discussed in this review were included if they explicitly reported vaccine-induced immune outcomes (e.g., antibody titers, seroconversion rates, or cellular responses) in relation to gut microbiome composition or function. Observational studies (cross-sectional, cohort) and interventional trials (antibiotic perturbation, probiotic or dietary interventions) were both considered, while purely animal or in vitro data were excluded from this section.

In Table 1, representative human studies linking gut microbiome composition or function to vaccine responsiveness are included.

### 3.1. Adults: Antibiotic Perturbation and Influenza Vaccines

The clearest demonstration of microbiome influence in adults comes from antibiotic-perturbation studies. In a randomized trial, Hagan et al. administered a five-day course of broad-spectrum antibiotics (neomycin, vancomycin, metronidazole) to healthy adults before inactivated influenza vaccination [9]. Antibiotic-treated participants exhibited profound reductions in bacterial load and diversity. While geometric-mean titers did not differ significantly overall, subgroup analysis revealed that individuals with low baseline influenza-specific antibody titers mounted significantly weaker responses after antibiotic depletion. Systems-level analyses further revealed altered transcriptional responses, metabolomic shifts (notably in bile-acid and SCFA pathways), and disrupted gut microbial ecology.

These findings highlight two important principles. First, microbiome effects on vaccine outcomes may be modulated by pre-existing immunity—more pronounced in primary responses than in booster contexts. Second, systemic metabolomic changes following microbiome perturbation suggest mechanistic plausibility for immune modulation. Similar context-dependent effects have been suggested for other vaccines, though comprehensive studies are lacking.

This subsection emphasizes randomized interventional trials (e.g., antibiotic challenge) as primary evidence for causal inference, while smaller observational cohorts were referenced to corroborate consistency across populations.

Observational studies in adults corroborate these findings. Microbiota composition, including enrichment of Prevotella and depletion of Bacteroides, has been associated with differential antibody titers to influenza vaccination [30]. In older adults, where immunosenescence diminishes vaccine responses, dysbiosis characterized by reduced diversity and increased pathobionts correlates with weaker responses to influenza and pneumococcal vaccines [31]. These associations suggest that microbiome state could serve as a biomarker of responsiveness in older populations.

### 3.2. Early Life: Neonatal Antibiotics, Bifidobacterium, and Routine Vaccines

Early life is a critical window for both microbiome assembly and immune-system development. Several birth cohorts across different geographies have consistently found that Bifidobacterium dominance in the first months of life correlates with stronger responses to routine vaccines. For example, infants with high Bifidobacterium abundance showed enhanced responses to BCG, oral polio, and pneumococcal vaccines [10,11]. Bifidobacteria promote immune tolerance and support Tfh and B-cell function, possibly via acetate production and epithelial conditioning.

The strongest evidence emerged recently from a prospective study by Ryan et al. (2025), which combined human cohorts with gnotobiotic mouse models [12]. Neonates exposed to antibiotics in the first week of life exhibited reduced antibody titers to multiple infant vaccines, including pneumococcal, Hib, and tetanus toxoid, persisting into the second year of life. These infants also demonstrated depletion of Bifidobacterium species. In mice, direct neonatal antibiotic exposure similarly blunted vaccine responses, but colonization with a defined Bifidobacterium consortium restored immunogenicity. This study provided causal evidence and positioned neonatal antibiotic exposure as a modifiable risk factor for poor vaccine outcomes.

The implications are profound: although antibiotics are often lifesaving in neonatal care, unnecessary or broad-spectrum use may have long-term consequences for vaccine-mediated protection. These data also highlight Bifidobacterium supplementation as a potential adjunctive strategy, though clinical trials are required. Inclusion here was restricted to studies reporting direct vaccine-outcome measures in infants, distinguishing them from correlational microbiome-development papers without immunogenicity endpoints.

### 3.3. Oral Vaccines and the “LMIC Efficacy Gap”

Oral vaccines (rotavirus, polio, cholera, typhoid) display well-recognized efficacy gaps between LMICs and HICs. Rotavirus vaccines, for example, demonstrate 85–95% efficacy in high-income settings but only 40–60% in many LMICs [12,13]. This discrepancy undermines disease-control efforts and has spurred investigation into microbiome contributions.

Multiple studies in South Asia and sub-Saharan Africa have correlated infant gut-microbiota composition with rotavirus-vaccine immunogenicity. In Bangladesh, responders exhibited higher relative abundance of Bifidobacterium longum and Enterococcus faecalis, whereas non-responders showed enrichment of Proteobacteria [32]. In Ghana, differences in microbial functional pathways rather than taxonomic composition predicted seroconversion [33]. A study in Malawi linked rotavirus-vaccine shedding and seroconversion to overall microbial diversity and enteropathogen burden, suggesting that high pathogen exposure and enteropathy alter the “microbial ecosystem” in ways detrimental to vaccine take [28].

Recent evidence has reinforced these associations. Wagner et al. (2025) examined the neonatal rotavirus vaccine RV3-BB in Indonesia and found that positive vaccine responses were associated with early-life microbiome features, including higher Bifidobacterium abundance and specific metabolic pathways [14]. These findings imply that targeted microbiome interventions could help close the oral-vaccine efficacy gap.

Similar patterns are seen for other oral vaccines. Polio-vaccine seroconversion has been linked with higher abundance of Bifidobacterium and lower prevalence of enteropathogenic *E. coli* [27]. For cholera and oral typhoid vaccines, baseline microbiota diversity and SCFA profiles correlate with immune outcomes, although data remain sparse [34]. Collectively, these studies suggest that microbiome features act as both biomarkers and potential modulators of oral-vaccine efficacy.

Inclusion criteria prioritized prospective birth cohorts and vaccine-trial sub-studies that measured both microbiome data and vaccine-immunogenicity outcomes, excluding ecological analyses lacking individual-level serologic endpoints.

### 3.4. Beyond Infants and Adults: Adolescents and Special Groups

Adolescents and young adults have been less studied, though emerging work in COVID-19 vaccine cohorts has suggested associations between gut-microbiome composition and antibody titers [35]. In immunocompromised populations, such as people living with HIV (PLWH), dysbiosis has been associated with reduced vaccine responses, including to COVID-19 vaccines [36]. These findings align with the concept that microbiome-mediated immune dysregulation could exacerbate pre-existing vulnerabilities.

Pregnant individuals represent another underexplored group. Maternal microbiome states may influence not only their own vaccine responses (e.g., to pertussis or influenza) but also the transfer of antibodies to the fetus via the placenta and breast milk. Preliminary studies suggest that maternal Bifidobacterium and SCFA profiles may condition neonatal immunity, though definitive evidence is lacking [29].

Where possible, only studies with defined human cohorts and reported vaccine-related immune endpoints were summarized, ensuring consistency of inclusion across age and physiological groups.

### 3.5. Critical Discussion and Emerging Concepts

Although evidence from human studies is growing, substantial variability remains. Differences in microbiota diversity, host genetics, diet, and environmental exposures can confound associations with vaccine outcomes. Study-design factors—such as small sample sizes, cross-sectional analyses, and differing microbiome sequencing methods—also complicate interpretation. To enhance reproducibility, future work should clearly delineate observational versus interventional designs and predefine microbiome-related inclusion/exclusion criteria (e.g., antibiotic exposure, probiotic use, comorbidities) to improve comparability across cohorts. The inclusion of diverse dietary patterns, antibiotic histories, and non-nutritive additives (e.g., artificial sweeteners) in study analyses could enhance reproducibility and mechanistic understanding.

Importantly, microbiome-targeted interventions are being explored not only in vaccine research but also in the modulation of immune-mediated diseases such as inflammatory bowel disease (IBD), reflecting the broader immunomodulatory potential of microbial manipulation [37]. This overlap strengthens the translational rationale for microbiome modulation as a general immunotherapeutic strategy.

Table 2 compares microbiome–vaccine associations across studies.

## 4. What Features of the Microbiome Matter?

While early studies emphasized the presence or absence of particular microbial taxa, it has become increasingly clear that the immunomodulatory influence of the gut microbiome on vaccine responses is multifactorial. Taxonomic composition, microbial diversity, functional capacity, and metabolite profiles all contribute, often interacting in complex ways. Understanding which features matter most is essential for developing predictive biomarkers and designing rational interventions.

### 4.1. Taxonomic Signatures

#### 4.1.1. Bifidobacterium as a Recurring Positive Correlate

Across multiple birth cohorts and vaccine types, Bifidobacterium spp. emerge as consistent positive correlates of immunogenicity. In Bangladesh and India, higher Bifidobacterium longum abundance during early infancy correlated with stronger responses to BCG, polio, and tetanus vaccines [10,11]. In a recent prospective study, depletion of Bifidobacterium following neonatal antibiotics was directly linked to weaker vaccine responses, and supplementation with Bifidobacterium restored immunogenicity in gnotobiotic mice [12]. These data provide both observational and causal support for a beneficial role.

Mechanistically, Bifidobacterium produce acetate, which strengthens epithelial barrier function, and exert immunomodulatory effects through exopolysaccharides and tryptophan metabolites. They also interact with host dendritic cells to promote tolerogenic but competent antigen presentation. Their dominance in breastfed infants, driven by human milk oligosaccharides (HMOs), may help explain why breastfed infants often show enhanced vaccine responsiveness relative to formula-fed peers.

#### 4.1.2. Other Beneficial Taxa

Beyond Bifidobacterium, certain Lactobacillus species, Faecalibacterium prausnitzii, and Akkermansia muciniphila have been associated with enhanced responses in some settings [38]. These taxa are SCFA producers and support epithelial health. In influenza vaccine studies, higher abundance of Faecalibacterium correlated with robust humoral responses [2]. However, these associations are less reproducible than those for Bifidobacterium.

#### 4.1.3. Detrimental Associations: Proteobacteria and Pathobionts

Conversely, enrichment of Proteobacteria (notably Escherichia coli pathotypes, Klebsiella, Enterobacter) is often linked to poorer vaccine responses, especially in LMIC settings where enteropathogen carriage is common [32,33]. High prevalence of enteropathogens not only alters microbial ecology but also induces chronic intestinal inflammation (“environmental enteric dysfunction”), which can blunt oral vaccine immunogenicity [14]. These findings suggest that microbiome composition can reflect both beneficial and detrimental ecological states relevant to vaccines.

### 4.2. Microbial Diversity and Stability

Diversity metrics provide another lens. In general, extremely low diversity, such as that induced by broad-spectrum antibiotics, correlates with weaker vaccine responses [9]. This may reflect the loss of functional redundancy and immunostimulatory cues. However, higher diversity is not always uniformly beneficial. In infants, early-life dominance by a few beneficial taxa (e.g., Bifidobacterium) may be more important than maximal diversity [27]. Thus, the relationship appears to follow a developmental trajectory: low but beneficially skewed diversity in early life, expanding to balanced diversity in adulthood.

Microbiome stability also matters. Longitudinal studies show that infants with more stable microbiota trajectories, even if low in diversity, are more likely to mount consistent vaccine responses [13]. Instability or dysbiosis, especially after antibiotics, may disrupt immune conditioning during critical windows.

### 4.3. Functional Capacity and Metagenomic Signatures

Taxonomic composition provides limited resolution, as different microbes can perform overlapping functions. Metagenomic and metatranscriptomic approaches have therefore been applied to vaccine studies, revealing that functional pathways may predict vaccine outcomes more reliably than taxa alone.

For example, Ghanaian infants who seroconverted after rotavirus vaccination exhibited enrichment of microbial genes involved in amino acid and vitamin biosynthesis [33]. In Bangladeshi infants, pathways related to carbohydrate metabolism and SCFA production correlated with positive responses [33]. In influenza vaccine cohorts, bile acid metabolism pathways predicted antibody titers better than taxonomic data [9].

These findings argue for a shift toward functional microbiomics, integrating gene content, transcriptional activity, and metabolite profiles, rather than focusing solely on taxonomy.

### 4.4. Metabolomic Profiles as Biomarkers

Metabolomics captures the downstream biochemical footprint of the microbiome and may offer the most direct readout of immune-relevant activity. Several metabolite classes have emerged as potential biomarkers:SCFAs: Higher baseline acetate and butyrate levels correlate with stronger antibody titers in some studies, though results vary depending on dose and context [20,21].Bile acids: Specific conjugated bile acids have been associated with impaired influenza vaccine responses, whereas balanced bile acid pools support Tfh differentiation [17].Indoles and tryptophan derivatives: Positive correlates of mucosal IgA responses and barrier function [9].Polyamines (spermidine, putrescine): Linked to enhanced autophagy and improved memory T-cell formation in preclinical models [8].

Metabolomic markers may also integrate environmental influences such as diet and antibiotic exposure, making them attractive for precision prediction.

### 4.5. Multi-Omics Integration

The complexity of microbiome–immune interactions has spurred the adoption of multi-omics approaches, combining metagenomics, metabolomics, transcriptomics, and systems serology. Integrative studies reveal that baseline microbial and metabolic states can explain a significant fraction of variance in vaccine responses. For example, systems-vaccinology analyses have demonstrated that microbiome-derived bile acid profiles interact with host transcriptional modules to shape influenza vaccine outcomes [9]. Similarly, combined microbiome–serology models improve prediction of rotavirus vaccine seroconversion compared with either data type alone [39].

These integrative approaches may yield predictive biomarkers capable of identifying individuals at risk for poor vaccine responses, enabling targeted interventions. They also provide mechanistic insight into how specific microbial functions translate into immune outcomes.

### 4.6. Developmental and Geographic Considerations

Which features matter most may depend on developmental stage and geography. In neonates, Bifidobacterium dominance and simple functional repertoires appear most important. In older children and adults, broader diversity and metabolic flexibility matter. In LMIC settings, pathogen burden and inflammatory milieu may overshadow beneficial features. Therefore, context-specific biomarkers are likely required, rather than a one-size-fits-all model.

## 5. Clinical Modulation Strategies

If the gut microbiome shapes vaccine outcomes, the next question is whether it can be deliberately manipulated to enhance immunogenicity. Multiple strategies have been explored, ranging from broadly accessible dietary interventions to targeted microbial products and adjuvants inspired by commensal signaling. Evidence remains mixed, but growing mechanistic and translational data point to several promising approaches.

### 5.1. Dietary Interventions and Prebiotics

Diet is the primary determinant of microbiome composition and function. Diets rich in complex carbohydrates and fermentable fibers increase short-chain fatty acid (SCFA) production, while Western-style diets low in fiber reduce microbial diversity and resilience. Given the immunomodulatory roles of SCFAs, dietary interventions are a logical means of supporting vaccine responses.

#### 5.1.1. Fiber Supplementation

Animal models provide proof-of-concept. Mice fed high-fiber diets exhibit enhanced antibody titers after influenza vaccination compared with low-fiber controls, an effect mediated by SCFAs that promote B cell differentiation and plasma-cell longevity [19]. SCFA supplementation restored impaired rabies-vaccine responses in antibiotic-treated mice [17].

Human data are less developed but encouraging. In small pilot trials, high-fiber diets or supplementation with inulin-type fructans have increased SCFA levels and modulated systemic immune signatures [30]. However, whether these changes translate into improved vaccine responses in humans has not yet been robustly demonstrated.

#### 5.1.2. Prebiotics and Human Milk Oligosaccharides (HMOs)

Prebiotics are nondigestible substrates that selectively stimulate beneficial microbes. In infants, prebiotics such as galacto-oligosaccharides (GOS) and fructo-oligosaccharides (FOS) promote Bifidobacterium dominance, paralleling the effects of HMOs in breast milk. Prebiotic supplementation could therefore indirectly enhance vaccine responsiveness by creating a favorable microbial ecology [32]. Randomized trials combining prebiotics with vaccines are limited, but feasibility is high, particularly in infant nutrition programs.

### 5.2. Probiotics and Synbiotics

Probiotics—live microorganisms that confer health benefits—are the most widely studied microbiome interventions in vaccinology. Synbiotics combine probiotics with prebiotics to support colonization and activity.

#### 5.2.1. Evidence from Randomized Trials

A 2024 systematic review and meta-analysis of probiotic supplementation and vaccine responses reported modest benefits, with significant heterogeneity [40]. Some trials found improved antibody titers to influenza, polio, and rotavirus vaccines, while others showed no effect. Variability in strain selection, dosing, duration, host population, and vaccine type complicates interpretation.

#### 5.2.2. The Case for Bifidobacterium

The strongest rationale exists for Bifidobacterium supplementation in infants, especially after antibiotic exposure. As noted in Section 3, neonatal antibiotics blunt vaccine responses, while Bifidobacterium consortia can restore them in preclinical models [12]. Several clinical trials are underway to test targeted Bifidobacterium strains (e.g., B. infantis EVC001) as adjuncts to routine immunization. If successful, this approach could be incorporated into neonatal care and immunization programs, especially in LMICs where antibiotic exposure is frequent.

#### 5.2.3. Safety and Regulatory Considerations

Probiotics are generally safe in healthy populations, but caution is warranted in immunocompromised individuals, preterm infants, and critically ill patients, where cases of bacteremia have been reported [41]. Regulatory frameworks for probiotics as vaccine adjuncts remain underdeveloped, requiring standardized quality control and strain documentation.

### 5.3. Postbiotics and Microbial Metabolites

Postbiotics—defined microbial products such as metabolites, cell-wall components, or lysates—offer an alternative strategy that avoids the uncertainties of colonization.

#### 5.3.1. SCFA Analogs and Delivery Systems

SCFAs are promising candidates but suffer from rapid absorption and systemic clearance. Synthetic SCFA analogs with improved pharmacokinetics are under development for inflammatory diseases and could be repurposed as vaccine adjuncts [42]. Microencapsulation technologies may allow targeted delivery of SCFAs to the gut, optimizing local immune conditioning before oral or parenteral vaccination.

#### 5.3.2. Indole Derivatives and AhR Ligands

Indole derivatives from tryptophan metabolism enhance mucosal IgA responses in preclinical models [9]. Purified or synthetic AhR ligands could be tested as adjuncts for oral vaccines, though safety and dosing require careful evaluation.

#### 5.3.3. Polyamines and Memory T Cell Support

Polyamines such as spermidine promote autophagy and long-lived memory T cell formation [8]. Early-phase clinical studies of spermidine supplementation in older adults have shown immune benefits, raising the possibility of use as a vaccine adjunct in immunosenescent populations [38].

### 5.4. Microbiome-Inspired Adjuvants

Another avenue is the rational design of vaccine adjuvants inspired by commensal signals.

#### 5.4.1. Flagellin and TLR5 Agonists

Bacterial flagellin, recognized by TLR5, acts as a potent adjuvant in animal models. Flagellin fusion proteins have enhanced antibody responses to influenza, plague, and other vaccines [18]. Structural engineering has yielded flagellin variants with reduced immunogenicity but preserved adjuvant activity, improving safety for clinical use [18].

#### 5.4.2. Other PRR Ligands

CpG oligodeoxynucleotides (TLR9 agonists) and monophosphoryl lipid A (TLR4 agonist) are already used as licensed adjuvants. While not microbiome derived per se, they exemplify the principle that microbial ligands can amplify vaccine responses. Future work may identify novel adjuvant candidates from commensal bacteria.

### 5.5. Fecal Microbiota Transplantation (FMT) and Next-Generation Approaches

Fecal microbiota transplantation (FMT) is an established therapy for recurrent Clostridioides difficile infection. Its potential role in vaccinology is speculative but conceptually intriguing. In theory, FMT could rapidly restore microbiome diversity and function after antibiotic disruption, possibly improving vaccine responses. However, safety, standardization, and acceptability challenges make FMT unlikely as a widespread vaccine adjunct.

More refined approaches are emerging, including defined microbial consortia, engineered probiotics, and microbiome-modulating small molecules. These “next-generation probiotics” may offer greater predictability and safety than traditional FMT or single-strain probiotics [38].

## 6. Special Populations

The influence of the gut microbiome on vaccine responses is not uniform across all individuals. Certain populations—because of their developmental stage, immune status, or environmental context—are particularly susceptible to microbiome-mediated modulation of immunogenicity. Understanding these groups is critical for targeted interventions and for advancing equitable vaccinology.

### 6.1. Neonates and Infants

#### 6.1.1. Early-Life Vulnerability

The neonatal period represents a window of both opportunity and vulnerability. The immune system is immature, characterized by skewing toward tolerance, reduced antigen-presenting capacity, and limited germinal center formation. At the same time, the microbiome is undergoing rapid assembly, strongly influenced by delivery mode, feeding type, antibiotic exposure, and environment.

Numerous studies show that infants with microbiota dominated by Bifidobacterium exhibit stronger responses to routine vaccines, including BCG, polio, and pneumococcal conjugates [10,11]. Conversely, neonatal antibiotic exposure—even when clinically necessary—can deplete Bifidobacterium and other beneficial taxa, leading to weaker responses persisting into the second year of life [12]. These findings suggest that safeguarding or restoring a healthy early microbiome is essential for optimal vaccine performance.

#### 6.1.2. Breastfeeding and Human Milk Oligosaccharides

Breast milk supports microbiome development through HMOs, which selectively enrich for Bifidobacterium. Breastfed infants not only display Bifidobacterium dominance but also have stronger antibody responses to certain vaccines compared with formula-fed infants [43]. Promoting breastfeeding thus has dual benefits: nutrition and microbiome-mediated immune support.

#### 6.1.3. Preterm Infants

Preterm infants face unique challenges. Their microbiomes are often shaped by prolonged hospitalization, antibiotic exposure, and formula feeding, resulting in dysbiosis characterized by low diversity and high Proteobacteria [19]. Immunogenicity of vaccines such as DTaP and Hib is already reduced in preterm populations, and microbiome disruption may exacerbate this. Probiotic supplementation has been explored in neonatal intensive care units, but its role in vaccine optimization remains to be established.

### 6.2. Older Adults and Immunosenescence

In older adults, immune aging (immunosenescence) and chronic low-grade inflammation (“inflammaging”) contribute to reduced vaccine efficacy. Influenza, pneumococcal, and herpes zoster vaccines all show diminished immunogenicity in this population [22].

The aging gut microbiome is typically less diverse, enriched in pathobionts, and depleted in beneficial SCFA producers [31]. Observational studies link these features to weaker influenza vaccine responses, though causality is difficult to establish [9]. Interventions such as dietary fiber supplementation or synbiotics may help restore microbial diversity and function, but controlled vaccine trials in older adults remain limited. Emerging evidence that polyamines such as spermidine support memory T cell formation [38] raises the possibility of targeted metabolic supplementation for this population.

### 6.3. Immunocompromised Populations

#### 6.3.1. People Living with HIV (PLWH)

PLWH often exhibit persistent gut dysbiosis characterized by reduced Bifidobacterium and Faecalibacterium and increased Prevotella. This dysbiosis correlates with impaired vaccine responses, including to COVID-19 vaccines [13]. While antiretroviral therapy restores some immune function, microbiome-targeted interventions may further improve vaccine performance in this group.

#### 6.3.2. Transplant Recipients and Patients on Immunosuppressive Therapy

In hematopoietic stem-cell-transplant (HSCT) recipients, microbiome diversity strongly predicts survival and infection risk. Vaccine responses post-transplant are typically poor, and antibiotics commonly used in conditioning regimens exacerbate dysbiosis. Probiotic or synbiotic supplementation is being cautiously explored, but safety remains paramount given risks of bacteremia.

### 6.4. Pregnant Individuals and Maternal–Infant Dyads

Maternal vaccination against influenza, pertussis, and COVID-19 provides critical protection for both mother and infant. Maternal microbiome composition may influence antibody titers and transplacental transfer, though data remain sparse. Breast-milk microbiota and metabolites further shape neonatal microbiome development and early immune priming [32]. Optimizing maternal microbiomes could thus indirectly enhance infant vaccine protection.

### 6.5. Low- and Middle-Income Countries (LMICs)

The greatest vaccine-efficacy gaps occur in LMICs, especially for oral vaccines. Environmental factors—malnutrition, high enteropathogen burden, poor sanitation—drive microbiome states associated with environmental enteric dysfunction (EED), a condition marked by chronic gut inflammation, barrier dysfunction, and impaired nutrient absorption [28].

EED is strongly linked to reduced oral-vaccine efficacy, particularly for rotavirus and polio. Interventions aimed at improving WASH (water, sanitation, hygiene) and nutrition are essential, but microbiome-targeted strategies may provide additional benefits. For example, synbiotics tailored to promote Bifidobacterium and SCFA production could be deployed alongside oral vaccines in LMIC settings [14].

## 7. Practical Guidance & Future Directions

The accumulating evidence linking the gut microbiome to vaccine responses raises critical questions for clinicians, researchers, and policymakers. How should microbiome variables be integrated into vaccine research and development? What practical steps can be taken in trial design? Which interventions are most promising for translation, and how can they be scaled equitably across diverse settings? This section outlines practical guidance and proposes future directions.

### 7.1. Integrating Microbiome Endpoints into Vaccine Trials

#### 7.1.1. Baseline Characterization

Future vaccine trials should systematically capture microbiome-related variables. At minimum, this includes dietary habits, recent or concurrent antibiotic and proton-pump-inhibitor (PPI) use, gastrointestinal symptoms, and breastfeeding status in infants. Collecting and biobanking stool samples enables later metagenomic, metabolomic, and multi-omics analyses. Such practices ensure that microbiome effects can be retrospectively interrogated, even if not part of the primary trial hypothesis.

Despite growing adoption of these practices, variability in collection timing, sequencing depth, and metadata completeness still hampers comparability across studies, highlighting the need for harmonized protocols.

#### 7.1.2. Pre-Existing Immunity

As demonstrated in adult influenza studies, microbiome influences are often strongest when baseline immunity is low [9]. Trials should measure and stratify participants by pre-existing antibody titers, prior vaccine doses, or exposure history. This will help distinguish microbiome effects on primary versus booster responses and prevent masking of associations.

#### 7.1.3. Longitudinal Sampling

Because vaccine responses evolve over time—priming, boosting, waning—longitudinal follow-up is essential. Trials should assess not only short-term antibody titers but also memory B/T cell responses, mucosal IgA, and durability of protection. Parallel microbiome sampling can identify temporal associations and causal dynamics.

However, existing longitudinal studies vary widely in follow-up duration and sampling frequency, leading to inconsistent detection of microbiome–immune correlations.

#### 7.1.4. Multi-Omics Integration

Combining microbiome data with systems vaccinology (transcriptomics, proteomics, metabolomics, systems serology) allows for more precise mechanistic insights. Integrated models can explain greater variance in vaccine outcomes than single-omics approaches [9,14]. Such designs also facilitate biomarker discovery for predictive or diagnostic use.

Yet, cross-cohort comparisons often yield conflicting molecular correlates—some identifying bile-acid pathways, others SCFA or indole signaling—reflecting both biological heterogeneity and methodological divergence. Consensus pipelines for data normalization and multi-omic integration are urgently needed.

### 7.2. Translational Priorities

#### 7.2.1. Targeted Interventions in Neonates and Infants

Given the strong evidence linking neonatal antibiotics and Bifidobacterium depletion to impaired vaccine responses [12], priority should be placed on interventions in early life. Trials could test defined Bifidobacterium consortia or synbiotics administered after antibiotic treatment, measuring impacts on both microbiome restoration and vaccine outcomes. If successful, such strategies could be incorporated into neonatal-care protocols worldwide.

Nevertheless, results across cohorts remain partly inconsistent: some studies in Asia show strong associations with Bifidobacterium longum, whereas European cohorts emphasize Bifidobacterium breve or Lactobacillus species, underscoring strain- and context-specific effects that require validation through harmonized multicenter RCTs.

#### 7.2.2. Optimizing Oral Vaccines in LMICs

Closing the efficacy gap for oral vaccines is a global-health imperative. Microbiome-targeted strategies—such as timed synbiotics, dietary supplementation, or microbial metabolites that enhance IgA responses—could be tested alongside rotavirus, polio, and cholera vaccination programs. Integration with WASH and nutritional interventions will be critical, as microbiome effects are embedded within broader environmental contexts [28].

Existing studies occasionally yield divergent predictors of vaccine “take,” ranging from diversity metrics to specific taxa, reflecting differences in sequencing depth and enteropathogen prevalence. Standardized endpoints and controlled interventional trials are needed to move beyond correlation.

Yet implementation in LMICs faces practical constraints: the cost of high-throughput sequencing, cold-chain requirements for probiotic products, and limited laboratory infrastructure often preclude large-scale microbiome monitoring. Strengthening local manufacturing of stable, lyophilized formulations and integrating microbiome endpoints into existing vaccine-trial platforms could improve feasibility. Cost-effectiveness analyses—comparing probiotic or dietary interventions with traditional adjuvant enhancements—will be essential to justify adoption. Furthermore, capacity-building partnerships with regional research institutes and ministries of health should accompany clinical trials to ensure sustainable translation rather than externally driven pilot projects.

#### 7.2.3. Older Adults and Immunosenescence

In aging populations, interventions such as dietary-fiber supplementation, synbiotics, or metabolic supplementation (e.g., spermidine) could be evaluated in the context of influenza, pneumococcal, and RSV vaccines [38]. These populations already receive high-dose or adjuvanted vaccines; microbiome-aware adjuncts could provide additional benefits.

To date, most studies are small pilot trials without control arms; large-scale randomized designs are required to determine causality and clinical benefit over existing vaccine enhancements.

#### 7.2.4. Immunocompromised Populations

For people living with HIV or transplant recipients, microbiome interventions could augment poor vaccine responses, but safety is paramount. Defined postbiotics or engineered probiotics with predictable profiles may be more appropriate than live microbial supplements. Rigorous phase I safety trials are needed before widespread implementation.

Future multicenter RCTs should stratify participants by immune status and immunosuppressive regimen to delineate safety and efficacy profiles, as findings remain inconsistent across small cohorts.

### 7.3. Research Gaps and Methodological Challenges

#### 7.3.1. Causality and Confounding

Most human studies remain observational, making causality difficult to prove. Future work should prioritize randomized controlled trials of microbiome-targeted interventions with vaccine immunogenicity as a primary endpoint. Cross-over designs, where feasible, could further strengthen causal inference.

In addition, adaptive-trial frameworks could test multiple probiotic or metabolite candidates simultaneously, accelerating discovery within 3–5 years. Without such designs, conflicting cohort findings will persist.

#### 7.3.2. Standardization of Methods

Microbiome research suffers from heterogeneity in sample collection, sequencing, bioinformatics pipelines, and statistical analysis. Standardized protocols are needed to ensure reproducibility and comparability across studies. Shared repositories and harmonized analytic frameworks would accelerate progress.

The lack of uniform metadata (diet, antibiotics, geography) explains much of the discordant evidence between cohorts, emphasizing the need for international guideline development akin to CONSORT standards for clinical trials.

#### 7.3.3. Geographic and Demographic Diversity

Most microbiome–vaccine studies to date have been conducted in North America, Europe, and select LMIC cohorts in South Asia and sub-Saharan Africa. Expanding research to under-represented regions and populations will provide a more comprehensive understanding and ensure that interventions are globally relevant.

Beyond geography, divergent dietary patterns, pathogen exposure, and host genetics contribute to conflicting outcomes across cohorts; coordinated multinational trials are essential to disentangle these variables.

#### 7.3.4. Beyond Bacteria: Virome and Mycobiome

While bacterial communities dominate current research, the gut virome (bacteriophages, enteric viruses) and mycobiome (fungi) may also shape immune responses. For example, fungal dysbiosis has been linked to altered vaccine outcomes in animal models, but human data are sparse. Incorporating these dimensions could yield new insights.

This represents a critical evidence gap where exploratory RCT sub-studies could quantify virome and mycobiome dynamics alongside bacterial and metabolomic endpoints.

### 7.4. Ethical and Regulatory Considerations

#### 7.4.1. Antibiotic Stewardship

The finding that neonatal antibiotics impair vaccine responses underscores the importance of antibiotic stewardship in early life. While antibiotics remain lifesaving, efforts should focus on minimizing unnecessary use, narrowing spectra, and promptly restoring microbiome health after treatment.

Strengthening evidence through pragmatic RCTs on microbiome restoration after antibiotic exposure will help define feasible clinical interventions.

#### 7.4.2. Regulatory Frameworks for Probiotics and Postbiotics

Probiotics and prebiotics are often marketed as dietary supplements with variable quality control. For use as vaccine adjuncts, regulatory standards for safety, efficacy, and strain documentation must be strengthened. Similarly, postbiotics and microbial metabolites may fall under novel therapeutic categories requiring new frameworks.

Coordination among regulatory agencies (FDA, EMA, WHO Prequalification) could facilitate consistent approval pathways for microbiome-based vaccine adjuncts within 5–10 years.

In practical terms, probiotics intended as vaccine adjuvants may need to transition from “food supplement” to “biologic adjunct” classification, necessitating adherence to GMP standards, stability testing, and clinical-trial oversight comparable to vaccines themselves. Postbiotics and purified microbial metabolites, by contrast, could be regulated under small-molecule or biologic-drug pathways, depending on their composition and mechanism. Early engagement with regulatory-science programs and harmonization across regions—particularly WHO’s Collaborative Registration Procedures and emerging African Medicines Agency frameworks—would expedite approval while ensuring safety and traceability.

#### 7.4.3. Equity and Access

Microbiome-targeted interventions must not exacerbate inequities. Strategies should prioritize affordability, scalability, and integration with existing immunization infrastructure, particularly in LMICs. Partnerships with local health systems and communities will be essential for equitable implementation.

Ensuring equitable trial representation and data sharing will also reduce bias and improve generalizability of causal evidence.

Beyond access, feasibility challenges remain: cold-chain maintenance for live probiotics, formulation stability under tropical conditions, and quality assurance during community-level distribution. Investment in thermostable formulations, community-based delivery through maternal-child-health programs, and tiered-pricing models could make microbiome-based adjuncts financially and logistically viable in resource-limited settings. Implementation research embedded in Gavi-supported vaccine programs could provide the necessary real-world evidence to inform sustainable scale-up.

### 7.5. Future Horizons

Looking ahead, several promising directions emerge:Predictive biomarkers: Development of microbiome–metabolite signatures that identify likely poor responders, enabling targeted interventions. Short-term (1–3 years) goals include harmonizing sample collection and sequencing standards across vaccine trials. Multicenter collaborations such as HIPC or the European Vaccine Initiative could integrate microbiome and metabolomic data to build predictive models validated in at least two independent populations.Personalized vaccinology: Tailoring vaccine formulations, adjuvants, or adjuncts based on an individual’s microbiome profile. Pilot interventional studies could test microbiome-informed vaccination schedules (e.g., probiotic preconditioning before influenza or rotavirus vaccination) within 3–5 years. Potential partners include national immunization programs, microbiome research networks, and precision-medicine consortia.Next-generation adjuvants: Rational design of commensal-derived molecules (e.g., flagellin derivatives, indole ligands) to boost immunogenicity safely. A 5–10-year research horizon should prioritize translational pipelines connecting microbial genomics, synthetic biology, and adjuvant chemistry. Collaborations between academic immunology labs, biotech startups, and public-private initiatives (e.g., CEPI, BARDA) could accelerate first-in-human trials.Integration with new vaccine platforms: Understanding how the microbiome influences responses to mRNA, nanoparticle, and mucosal vaccines, which may be more sensitive to metabolic and barrier contexts. Ongoing COVID-19 and RSV vaccine studies provide immediate opportunities for microbiome sub-analyses; large-scale consortia such as NIH’s PREVENT-19 or Horizon Europe programs could embed microbiome modules into platform-vaccine trials within the next 2–4 years.Systems vaccinology: Embedding microbiome analyses into the systems-vaccinology paradigm to generate holistic models of vaccine response. Future efforts (5–8 years) should integrate multi-omic layers—metagenomics, transcriptomics, metabolomics, and immunophenotyping—into unified computational frameworks. Collaborative modeling centers (e.g., Allen Institute for Immunology, Global Virome Project) could serve as hubs for cross-disciplinary data integration and predictive modeling.

## 8. Conclusions

Over the past decade, the gut microbiome has emerged as a critical determinant of vaccine immunogenicity. Mechanistic studies reveal that commensal microbes provide innate signals (e.g., flagellin via TLR5) and metabolic cues (e.g., SCFAs, bile acids, indole derivatives) that shape antigen presentation, germinal center reactions, and the durability of adaptive immunity. Human data reinforce these findings: in adults, antibiotic-induced depletion alters responses to influenza vaccines, particularly in those with low baseline immunity; in infants, Bifidobacterium dominance supports strong responses to multiple routine vaccines, whereas neonatal antibiotics diminish them; and in LMIC settings, oral vaccine underperformance correlates with microbiome composition, functional capacity, and enteropathogen burden.

Together, these observations highlight the microbiome as both a biomarker and a target for intervention. Modulation strategies range from dietary fiber and prebiotics that promote beneficial taxa, to probiotics and synbiotics (notably Bifidobacterium) aimed at restoring neonatal or antibiotic-disrupted microbiomes, to postbiotics and microbial metabolites that could provide defined, scalable adjuncts. Microbiome-inspired adjuvants such as engineered flagellin derivatives illustrate how endogenous microbial signals can be rationally harnessed. Although fecal microbiota transplantation and engineered consortia remain experimental, they underscore the potential of next-generation microbiome therapeutics.

Special populations—neonates, preterm infants, older adults, immunocompromised individuals, pregnant women, and LMIC communities—represent contexts where microbiome-aware vaccinology may yield the most substantial benefits. Embedding microbiome endpoints into vaccine trials, developing predictive biomarkers, and integrating microbiome research with systems vaccinology will be essential next steps. Equally important are ethical and regulatory frameworks that promote antibiotic stewardship, ensure safety and standardization of microbial interventions, and guarantee equitable access across settings.

Looking forward, the convergence of microbiome science, immunology, and vaccine technology creates opportunities to move from correlation to causation, from observation to intervention, and from population averages to precision approaches. By incorporating the microbiome into the design and evaluation of vaccines, we can aspire to enhance immunogenicity, extend durability, and reduce global disparities in vaccine performance. In this vision, microbiome-aware vaccinology represents not only a new scientific frontier but also a practical pathway toward universal, durable, and equitable protection against infectious diseases.

## Figures and Tables

**Table 1 vaccines-13-01116-t001:** Representative human studies linking gut microbiome composition or function to vaccine responsiveness.

Study Population	Vaccine Type	Key Microbial Correlates	Observed Outcome	References
Healthy adults (USA)	Inactivated influenza	Bacteroides depletion, reduced SCFAs after antibiotics	Diminished antibody response in those with low baseline titers	[9]
Infants (Bangladesh)	Oral polio	High Bifidobacterium and Lactobacillus abundance	Enhanced IgA and seroconversion	[27]
Neonates (Multicenter)	Pneumococcal, Hib, tetanus	Early-life antibiotics → reduced Bifidobacterium	Persistently lower antibody titers; restored by Bifidobacterium colonization	[12]
Infants (Malawi)	Rotavirus (oral)	Low diversity, high enteropathogen burden	Decreased vaccine take and seroconversion	[28]
Adults (China)	SARS-CoV-2 (mRNA)	Abundance of Bifidobacterium adolescentis; SCFA production	Higher neutralizing antibody titers	[24]
Older adults (UK)	Pneumococcal and influenza	Reduced diversity, enrichment of pathobionts	Poorer immunogenicity	[29]

**Table 2 vaccines-13-01116-t002:** Representative microbiome–vaccine associations across studies.

Vaccine Type	Population/Model	Key Microbiome Features	Proposed Mechanism	Direction of Effect	References
Influenza (inactivated)	Adults (antibiotic-treated)	↓ Diversity; loss of flagellated taxa	TLR5/flagellin–NF-κB signaling	↓ Antibody titers	[9]
Influenza (elderly)	Older adults	↑ Pathobionts; ↓ SCFA producers	Reduced Tfh function/immunosenescence	↓ Response	[23]
BCG/Polio/Pneumococcal	Infants	↑ *Bifidobacterium* dominance	SCFA & acetate → enhanced Th1/Th17 + IgA	↑ Response	[12]
Rotavirus (oral)	Infants (LMIC)	↑ *B. longum*/*E. faecalis*	Mucosal IgA enhancement	↑ Seroconversion	[13,29]
HBV	Adults	Distinct bile-acid profile	FXR/TGR5 signaling → Tfh differentiation	Variable	[17]
COVID-19	Adults	Gut dysbiosis; ↓ *Faecalibacterium*	Systemic cytokine modulation	↓ Response	[24,35]

## Data Availability

Data sharing is not applicable to this article as no new data were created or analyzed in this study. All data supporting the findings of this study are available within the cited references.

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
