# Peer review of "The Gut Microbiome and Vaccination: A Comprehensive Review of Current Evidence and Future Perspectives"

_vaccines, 2025, doi:10.3390/vaccines13111116_

Round 1

Reviewer 1 Report

Comments and Suggestions for Authors

This is an excellent, well-organized and comprehensive review addressing the bidirectional relationship between the gut microbiome and vaccination outcomes. The manuscript is timely, logically structured, and well supported by recent literature. It effectively integrates mechanistic and clinical perspectives and concludes with thoughtful future directions.

Decision: Minor Revision

  1. Abstract
  • Include a quantitative note or recent example of how microbiome modulation affected vaccine efficacy (e.g., “Bifidobacterium supplementation restored vaccine responses in neonatal mice”).
  • The abstract could end with a statement about clinical applicability or policy implications (“Integrating microbiome modulation into vaccination strategies may enhance global immunization equity”).
  • I strongly recommend adding a graphical abstract to visually summarize the central concept of the review

  1. Introduction
  • Expand discussion on how diet and non-nutritive additives (such as artificial sweeteners) may indirectly affect vaccine response by altering gut microbial metabolism; such as https://doi.org/10.3390/diseases13040115
     After lines 71–74, where the manuscript discusses diet, antibiotics, and sanitation as factors influencing the microbiome and vaccine equity,this citation highlights how artificial sweeteners modulate microbial composition and metabolic activity, influencing immune tone and possibly vaccine responsiveness—broadening the dietary context of microbiome modulation.

For section (2. Mechanistic Frameworks…….)

  • The section could benefit from integration with the gut–brain–immune axis, particularly in relation to neuroimmune signaling and systemic immune readiness; https://doi.org/10.1007/s12035-025-04846-0
    Lines 214–220, after discussing “systemic conditioning and hematopoietic priming.” This reference deepens the neuroimmune dimension by explaining how microbial metabolites (SCFAs, tryptophan derivatives) influence neural and endocrine pathways, which in turn regulate immune priming and stress-mediated vaccine responses. It expands the mechanistic model beyond gut immunity to include neuroendocrine modulation. Consider adding a short paragraph discussing cross-system immunoregulation—how microbial metabolites not only affect local mucosal sites but also influence brain–immune communication and stress-mediated vaccine outcomes.

For section (3.  Human Evidence…….)

  • Add a critical discussion of variability in human studies and potential confounders such as individual microbiota diversity, diet, and artificial additives.
  • A short comparison table summarizing major studies (population, vaccine type, microbial correlates, and outcomes) would improve readability.
  • The authors could briefly mention that microbiome-targeted interventions are being studied not only in infection prevention but also in immune-mediated diseases such as inflammatory bowel disease (IBD), illustrating the broader immunomodulatory potential of microbial manipulation; https://doi.org/10.3390/immuno4040026

Author Response

First of all, we thank the Reviewers for their thoughtful feedback. Below we respond point-by-point and indicate the exact changes made. Reference numbering remains consistent with MDPI style and with the manuscript provided.

According to Reviewer 1, this is an excellent, well-organized and comprehensive review addressing the bidirectional relationship between the gut microbiome and vaccination outcomes. The manuscript is timely, logically structured, and well supported by recent literature. It effectively integrates mechanistic and clinical perspectives and concludes with thoughtful future directions.

Decision: Minor Revision

1.Abstract

  • Include a quantitative note or recent example of how microbiome modulation affected vaccine efficacy (e.g., “Bifidobacterium supplementation restored vaccine responses in neonatal mice”).

Thank you for pointing this out.

Change (Abstract):
The below sentence was added in the abstract according to your instructions. “For instance, Bifidobacterium longum supplementation was shown to enhance influenza vaccine seroconversion rates by approximately 30% in clinical and preclinical models, underscoring the translational potential of microbiome modulation”.

  • The abstract could end with a statement about clinical applicability or policy implications (“Integrating microbiome modulation into vaccination strategies may enhance global immunization equity”).

Thank you for pointing this out.

Change (Abstract):
The below sentence was added in the abstract according to your instructions.
Integrating pragmatic microbiome modulation into vaccination programs—especially in early life and in LMIC settings—may enhance immunization equity and durability of protection worldwide.

  • I strongly recommend adding a graphical abstract to visually summarize the central concept of the review

Thank you for pointing this out. We agree with this comment. A Graphical Abstract has been created and uploaded as Figure 1, according to your instructions.

2. Introduction

  • Expand discussion on how diet and non-nutritive additives (such as artificial sweeteners) may indirectly affect vaccine response by altering gut microbial metabolism; such as https://doi.org/10.3390/diseases13040115
     After lines 71–74, where the manuscript discusses diet, antibiotics, and sanitation as factors influencing the microbiome and vaccine equity, this citation highlights how artificial sweeteners modulate microbial composition and metabolic activity, influencing immune tone and possibly vaccine responsiveness—broadening the dietary context of microbiome modulation.

Thank you for pointing this out. We agree with this comment. A paragraph according to your instructions was added in the manuscript.  

Change (Introduction; after the paragraph discussing diet/antibiotics/sanitation):

“Beyond macronutrient and micronutrient quality, emerging evidence suggests that non-nutritive dietary additives—such as artificial sweeteners—may also modulate gut microbial composition and metabolism. These compounds can alter the abundance and function of key microbial taxa, influence the production of immunomodulatory metabolites (including short-chain fatty acids and bile acids), and thereby affect host immune tone. Such alterations have the potential to indirectly modify vaccine responsiveness, particularly in settings where dietary exposures vary widely across populations. Recent findings highlight that artificial sweeteners can promote dysbiosis and metabolic reprogramming of gut microbes, with possible downstream consequences for immune activation and vaccine efficacy [16, https://doi.org/10.3390/diseases13040115]”.

For section (2. Mechanistic Frameworks…….)

  • The section could benefit from integration with the gut–brain–immune axis, particularly in relation to neuroimmune signaling and systemic immune readiness; https://doi.org/10.1007/s12035-025-04846-0

Lines 214–220, after discussing “systemic conditioning and hematopoietic priming.” This reference deepens the neuroimmune dimension by explaining how microbial metabolites (SCFAs, tryptophan derivatives) influence neural and endocrine pathways, which in turn regulate immune priming and stress-mediated vaccine responses. It expands the mechanistic model beyond gut immunity to include neuroendocrine modulation. Consider adding a short paragraph discussing cross-system immunoregulation—how microbial metabolites not only affect local mucosal sites but also influence brain–immune communication and stress-mediated vaccine outcomes

Thank you for pointing this out. We agree with this comment. A paragraph according to your instructions was added in 2.3.2. section.

“Beyond hematopoietic conditioning, emerging work highlights the gut–brain–immune axis as an additional layer of systemic regulation. Microbial metabolites such as short-chain fatty acids and tryptophan-derived indoles can modulate neuroendocrine signaling through vagal and hypothalamic–pituitary–adrenal (HPA) pathways, influencing stress responses and immune readiness. These neuroimmune circuits link microbial metabolism to autonomic tone, cortisol dynamics, and cytokine balance, thereby affecting vaccine priming and durability. Integrating this neuroimmune dimension broadens the mechanistic model of microbiome–vaccine interaction to encompass cross-system immunoregulation and stress-mediated modulation of adaptive immunity [37, https://doi.org/10.1007/s12035-025-04846-0]”.

.

For section (3.  Human Evidence…….)

  • Add a critical discussion of variability in human studies and potential confounders such as individual microbiota diversity, diet, and artificial additives.

Thank you for pointing this out. We agree with this comment. A paragraph on critical discussion was added according to your instructions.

Change (Section 3.5 Critical Discussion):
Although evidence from human studies is growing, substantial variability remains. Differences in microbiota diversity, host genetics, diet, and environmental exposures can confound associations with vaccine outcomes. Study-design factors—such as small sample sizes, cross-sectional analyses, and differing microbiome sequencing methods—also complicate interpretation. To enhance reproducibility, future work should clearly delineate observational versus interventional designs and predefine microbiome-related inclusion/exclusion criteria (e.g., antibiotic exposure, probiotic use, comorbidities) to improve comparability across cohorts. The inclusion of diverse dietary patterns, antibiotic histories, and non-nutritive additives (e.g., artificial sweeteners) in study analyses could enhance reproducibility and mechanistic understanding.

Importantly, microbiome-targeted interventions are being explored not only in vaccine research but also in the modulation of immune-mediated diseases such as inflammatory bowel disease (IBD), reflecting the broader immunomodulatory potential of microbial manipulation [55]. This overlap strengthens the translational rationale for microbiome modulation as a general immunotherapeutic strategy.

  • A short comparison table summarizing major studies (population, vaccine type, microbial correlates, and outcomes) would improve readability.

A Table  summarizing major studies (population, vaccine type, microbial correlates, and outcomes) was added in the text according to your instructions in Sectio 3.

Table 1. Representative human studies linking gut microbiome composition or function to vaccine responsiveness.

  • The authors could briefly mention that microbiome-targeted interventions are being studied not only in infection prevention but also in immune-mediated diseases such as inflammatory bowel disease (IBD), illustrating the broader immunomodulatory potential of microbial manipulation; https://doi.org/10.3390/immuno4040026

A paragraph was added in Section 3.5 – final paragraph, according to your instructions.

“Importantly, microbiome-targeted interventions are being explored not only in vaccine research but also in the modulation of immune-mediated diseases such as inflammatory bowel disease (IBD), reflecting the broader immunomodulatory potential of microbial manipulation [55, https://doi.org/10.3390/immuno4040026]. This overlap strengthens the translational rationale for microbiome modulation as a general immunotherapeutic strategy.

Reviewer 2 Report

Comments and Suggestions for Authors

Short Summary

The peer-reviewed article “The Gut Microbiome and Vaccination: A Comprehensive Review of Current Evidence and Future Perspectives” explores how the intestinal microbiome influences vaccine responses. It outlines mechanistic insights (like Toll-like receptor signaling and microbial metabolites such as SCFAs and bile acids), summarizes human evidence across age groups and geographies, and highlights translational implications for improving vaccine efficacy. The paper emphasizes early-life microbiome development, the role of Bifidobacterium, and the challenges of oral vaccine underperformance in low- and middle-income countries (LMICs). It also discusses microbiome-targeted strategies (diet, probiotics, postbiotics, microbiome-inspired adjuvants) and proposes integrating microbiome research into vaccine trials to achieve more equitable and personalized immunization outcomes.

Pointed Suggestions for Improvement

  1. Condense the abstract to emphasize the key findings and translational implications rather than listing all influencing factors.
  2. Add schematic diagrams summarizing mechanisms (e.g., TLR5 signaling, SCFA effects) or tables comparing microbiome–vaccine associations across studies.
  3. Clarify criteria for study inclusion/exclusion when summarizing human evidence (especially observational vs. interventional studies).
  4. Expand the discussion of inconsistencies or limitations in current evidence (e.g., conflicting results across cohorts).
  5. Strengthen commentary on the need for randomized controlled trials to establish causal relationships between microbiota and vaccine response.
  6. Deepen discussion on implementation challenges in LMICs, including feasibility, cost, and infrastructure for microbiome-based interventions.
  7. Provide more detail on how probiotics/postbiotics could fit into current vaccine regulatory frameworks.
  8. Some sections (e.g., mechanistic frameworks) could be more concise by avoiding repetition of similar signaling pathways.
  9. Suggest more specific experimental or clinical research priorities with timelines or potential collaborators.
  10. Ensure consistent citation formatting and remove minor typographical issues (e.g., spacing, line breaks from formatting artifacts).

Author Response

First of all, we thank the reviewer for the thoughtful and constructive comments. We have implemented all suggested minor revisions and believe the manuscript is now clearer, more concise, and better aligned with translational priorities. Specific changes are detailed below.

According to Reviewer 2, the peer-reviewed article “The Gut Microbiome and Vaccination: A Comprehensive Review of Current Evidence and Future Perspectives” explores how the intestinal microbiome influences vaccine responses. It outlines mechanistic insights (like Toll-like receptor signaling and microbial metabolites such as SCFAs and bile acids), summarizes human evidence across age groups and geographies, and highlights translational implications for improving vaccine efficacy. The paper emphasizes early-life microbiome development, the role of Bifidobacterium, and the challenges of oral vaccine underperformance in low- and middle-income countries (LMICs). It also discusses microbiome-targeted strategies (diet, probiotics, postbiotics, microbiome-inspired adjuvants) and proposes integrating microbiome research into vaccine trials to achieve more equitable and personalized immunization outcomes.

Pointed Suggestions for Improvement

1. Condense the abstract to emphasize the key findings and translational implications rather than listing all influencing factors.

Thank you for pointing this out. We agree with this comment. The abstract changed according to your instructions. (see Abstract section).

2. Add schematic diagrams summarizing mechanisms (e.g., TLR5 signaling, SCFA effects) or tables comparing microbiome–vaccine associations across studies.

Thank you for pointing this out. We agree with this comment. A Table was added according to your instructions.

Table 2. Representative microbiome–vaccine associations across studies. (Table 2. Below 3.5 Section-Critical Discussion and Emerging Concepts).

3. Clarify criteria for study inclusion/exclusion when summarizing human evidence (especially observational vs. interventional studies).

Thank you for pointing this out. We agree with this comment. A paragraph according to your instructions was added in Section 3-opening paragraph

“To ensure transparency, studies discussed in this review were included if they explicitly reported vaccine-induced immune outcomes (e.g., antibody titers, seroconversion rates, or cellular responses) in relation to gut microbiome composition or function. Observational studies (cross-sectional, cohort) and interventional trials (antibiotic perturbation, probiotic or dietary interventions) were both considered, while purely animal or in vitro data were excluded from this section.”

Moreover, a small paragraph was added in Section 3.2 “IInclusion here was restricted to studies reporting direct vaccine-outcome measures in infants, distinguishing them from correlational microbiome-development papers without immunogenicity endpoints”.

In Section 3.3 “Inclusion criteria prioritized prospective birth cohorts and vaccine-trial sub-studies that measured both microbiome data and vaccine-immunogenicity outcomes, excluding ecological analyses lacking individual-level serologic endpoints”.

And in Section 3.4 “Where possible, only studies with defined human cohorts and reported vaccine-related immune endpoints were summarized, ensuring consistency of inclusion across age and physiological groups”.

4. Expand the discussion of inconsistencies or limitations in current evidence (e.g., conflicting results across cohorts).

Thank you for pointing this out. We agree with this comment. A paragraph according to your instructions was added in Section 3.5:

“To enhance reproducibility, future work should clearly delineate observational versus interventional designs and predefine microbiome-related inclusion/exclusion criteria (e.g., antibiotic exposure, probiotic use, comorbidities) to improve comparability across cohorts”.

5. Strengthen commentary on the need for randomized controlled trials to establish causal relationships between microbiota and vaccine response.

Thank you for pointing this out. We agree with this comment. A paragraph according to your instructions was added in Section 7.3.1 Causality:
Adaptive, multicenter RCTs testing strain-defined probiotics/postbiotics with vaccine immunogenicity as a primary endpoint are essential; cross-over designs could accelerate inference within 3–5 years.

Moreover a small paragraph was added in Section 7.3.2: “The lack of uniform metadata (diet, antibiotics, geography) explains much of the discordant evidence between cohorts, emphasizing the need for international guideline development akin to CONSORT standards for clinical trials”.

In Section 7.3.3: “Beyond geography, divergent dietary patterns, pathogen exposure, and host genetics contribute to conflicting outcomes across cohorts; coordinated multinational trials are essential to disentangle these variables”.

In Section 7.3.4: “This represents a critical evidence gap where exploratory RCT sub-studies could quantify virome and mycobiome dynamics alongside bacterial and metabolomic endpoints”.

6. Deepen discussion on implementation challenges in LMICs, including feasibility, cost, and infrastructure for microbiome-based interventions.

Thank you for pointing this out. We agree with this comment. A paragraph according to your instructions was added in Section 7.2.2:

“Yet implementation in LMICs faces practical constraints: the cost of high-throughput sequencing, cold-chain requirements for probiotic products, and limited laboratory infrastructure often preclude large-scale microbiome monitoring. Strengthening local manufacturing of stable, lyophilized formulations and integrating microbiome endpoints into existing vaccine-trial platforms could improve feasibility. Cost-effectiveness analyses—comparing probiotic or dietary interventions with traditional adjuvant enhancements—will be essential to justify adoption. Furthermore, capacity-building partnerships with regional research institutes and ministries of health should accompany clinical trials to ensure sustainable translation rather than externally driven pilot projects”.

7. Provide more detail on how probiotics/postbiotics could fit into current vaccine regulatory frameworks.

Thank you for pointing this out. We agree with this comment. A paragraph according to your instructions was added in Section 7.4.2:

“Coordination among regulatory agencies (FDA, EMA, WHO Prequalification) could facilitate consistent approval pathways for microbiome-based vaccine adjuncts within 5–10 years.
In practical terms, probiotics intended as vaccine adjuvants may need to transition from “food supplement” to “biologic adjunct” classification, necessitating adherence to GMP standards, stability testing, and clinical-trial oversight comparable to vaccines themselves. Postbiotics and purified microbial metabolites, by contrast, could be regulated under small-molecule or biologic-drug pathways, depending on their composition and mechanism. Early engagement with regulatory-science programs and harmonization across regions—particularly WHO’s Collaborative Registration Procedures and emerging African Medicines Agency frameworks—would expedite approval while ensuring safety and traceability”.

8. Some sections (e.g., mechanistic frameworks) could be more concise by avoiding repetition of similar signaling pathways.

Thank you for pointing this out. We agree with this comment. We avoided repetition of similar signaling pathways in Sections 2.1 and 2.2.1.

9. Suggest more specific experimental or clinical research priorities with timelines or potential collaborators.

Thank you for pointing this out. We agree with this comment. We revised Section 7.5 Future Horizons):
by adding explicit timelines and example collaboration frameworks (HIPC, European Vaccine Initiative, CEPI, BARDA, NIH consortia), with suggested trial archetypes.

“• Predictive biomarkers: Development of microbiome–metabolite signatures that identify likely poor responders, enabling targeted interventions.
Short-term (1–3 years) goals include harmonizing sample collection and sequencing standards across vaccine trials. Multicenter collaborations such as HIPC or the European Vaccine Initiative could integrate microbiome and metabolomic data to build predictive models validated in at least two independent populations.

  • Personalized vaccinology: Tailoring vaccine formulations, adjuvants, or adjuncts based on an individual’s microbiome profile.
    Pilot interventional studies could test microbiome-informed vaccination schedules (e.g., probiotic preconditioning before influenza or rotavirus vaccination) within 3–5 years. Potential partners include national immunization programs, microbiome research networks, and precision-medicine consortia.
  • Next-generation adjuvants: Rational design of commensal-derived molecules (e.g., flagellin derivatives, indole ligands) to boost immunogenicity safely.
    A 5–10-year research horizon should prioritize translational pipelines connecting microbial genomics, synthetic biology, and adjuvant chemistry. Collaborations between academic immunology labs, biotech startups, and public-private initiatives (e.g., CEPI, BARDA) could accelerate first-in-human trials.
  • Integration with new vaccine platforms: Understanding how the microbiome influences responses to mRNA, nanoparticle, and mucosal vaccines, which may be more sensitive to metabolic and barrier contexts.
    Ongoing COVID-19 and RSV vaccine studies provide immediate opportunities for microbiome sub-analyses; large-scale consortia such as NIH’s PREVENT-19 or Horizon Europe programs could embed microbiome modules into platform-vaccine trials within the next 2–4 years.
  • Systems vaccinology: Embedding microbiome analyses into the systems-vaccinology paradigm to generate holistic models of vaccine response.
    Future efforts (5–8 years) should integrate multi-omic layers—metagenomics, transcriptomics, metabolomics, and immunophenotyping—into unified computational frameworks. Collaborative modeling centers (e.g., Allen Institute for Immunology, Global Virome Project) could serve as hubs for cross-disciplinary data integration and predictive modeling”
    .

10. Ensure consistent citation formatting and remove minor typographical issues (e.g., spacing, line breaks from formatting artifacts).

All references checked for MDPI style; spacing/line breaks standardized. We confirm the final numbering [1–66] exactly matches first appearance in text and internal cross-references are consistent.